# Assessing Affective State in Laboratory Rodents to Promote Animal Welfare—What Is the Progress in Applied Refinement Research?

**DOI:** 10.3390/ani9121026

**Published:** 2019-11-25

**Authors:** Paulin Jirkof, Juliane Rudeck, Lars Lewejohann

**Affiliations:** 1Department Animal Welfare and 3R, University of Zurich, 8057 Zurich, Switzerland; 2German Federal Institute for Risk Assessment (BfR), German Center for the Protection of Laboratory Animals (Bf3R), 12277 Berlin, Germany; Juliane.Rudeck@bfr.bund.de (J.R.); Lars.Lewejohann@bfr.bund.de (L.L.); 3Institute of Animal Welfare, Animal Behavior and Laboratory Animal Science, Freie University Berlin, 14163 Berlin, Germany

**Keywords:** refinement, animal welfare, affective state, rodents, animal experiment

## Abstract

**Simple Summary:**

In the past, there was a strong focus on avoiding or reducing negative animal welfare in animal experimentation. Recently, the importance of promoting positive animal welfare in laboratory animals has been highlighted. To ensure and promote positive animal welfare, reliable methods to evaluate the animal’s emotional state are important. Important achievements have been made to assess pain and other negative states in animals in the last decades, and only recently have positive emotions been gaining more interest. Therefore, more methods allowing the assessment of emotional states in animals have been introduced. In this overview article, we present common and emerging methods to assess emotions in laboratory rodents. We focus on the use of these methods in applied refinement research to identify achieved progress as well as the potential of these tools to improve animal welfare in animal-based research and animal experimentation.

**Abstract:**

An animal’s capacity to suffer is a prerequisite for any animal welfare concern, and the minimization of suffering is a key aim of refinement research. In contrast to the traditional focus on avoiding or reducing negative welfare states, modern animal welfare concepts highlight the importance of promoting positive welfare states in laboratory animals. Reliable assessments of affective states, as well as the knowledge of how to elicit positive affective states, are central to this concept. Important achievements have been made to assess pain and other negative affective states in animals in the last decades, but it is only recently that the neurobiology of positive emotions in humans and animals has been gaining more interest. Thereby, the need for promotion of positive affective states for laboratory animals is gaining more acceptance, and methods allowing the assessment of affective states in animals have been increasingly introduced. In this overview article, we present common and emerging methods to assess affective states in laboratory rodents. We focus on the implementation of these methods into applied refinement research to identify achieved progress as well as the future potential of these tools to improve animal welfare in animal-based research.

## 1. Introduction

Although the number of animals used for research purposes is small compared to animals used for food production, scientists in laboratory animal science are under particular pressure not only to justify the use of animals but also to promote animal welfare.

The concept of animal welfare has evolved rapidly over the last 60 years [1,2,3,4,5,6] and comprises provisions and freedoms that can be summarized as: (1) provision of food and water to maintain health and vigor, (2) freedom from discomfort by providing appropriate living conditions, (3) freedom from pain, injury, and disease including appropriate health care, (4) freedom from fear and distress by avoiding conditions eliciting mental suffering, and (5) provision of social and physical environment to perform important species typical behaviors. Notably, it has been acknowledged that above merely avoiding unfavorable conditions, animal welfare also comprises activities eliciting affective states related to a good living [3,7,8]. This, however, is not always easy to be achieved in the context of animal experimentation and might be subjected to compromises related to the purpose of the human custody. Moreover, when integrating over the lifetime of an animal, it is not reasonable to expect that any individual maintains a constant level of welfare throughout its entire life. Therefore, it should be considered not only to reduce events associated with poor animal welfare but also to introduce situations eliciting good animal welfare in order to promote a life worth living.

In order to be able to make statements about the state of well-being at all, a reliable spectrum of methods is required for the analysis of the current state of an animal. Aside from objective measurements, as for example, health status, reproductive success, or lifespan, the core unit of animal welfare is the subjective perception of an individual. From an animal’s point of view, a certain situation can be perceived as negative or even life-threatening, even if it is not objectively so. This reflects a differentiation between canonical and perceived costs, but both must not be neglected [9]. The individual perception of the current welfare state is related to concepts such as “Feelings”, “Emotions”, “Sentiment”, or “Mood”. We are aware that all these terms have a strong anthropomorphic connotation, which complicates an objective scientific discussion of the subject. Although not totally unbiased, we will use the term “affective state” for referring to the animal’s state in relation to subjective perceptions of internal or external stimuli.

Affective states are defined as multifaceted phenomena with neuronal, physiological, behavioral, cognitive, and subjective aspects and can have different levels of arousal and either positive or negative valence [10,11]. Affective states are heavily involved in motivating behavior [12] and, therefore, can be seen as an evolutionary adaptation [13].

Today, it is assumed that all mammals, and probably all vertebrates, have at least some capacity for primal affective states [14,15,16]. Individual perception seems to notoriously elude scientific quantification and, especially in animals, it is a great challenge to measure affective states. However, considerable progress has recently been made in theoretical concepts and methods that increasingly allow affective states to be quantified [17,18,19,20].

Affective states with their positive or negative valence have the potential to reflect enhanced or compromised overall quality of life within a certain time frame and are, therefore, important concepts for animal welfare and refinement research. The importance of the detection and avoidance of negative affective states is, therefore, long established in the laboratory animal science community. The use of appropriate anesthesia and analgesia or the implementation of less invasive methods and other stress-reducing measures are important tools to minimize the occurrence of negative affective states like pain or distress in laboratory animals. Recently, there has been a growing awareness of the need to not only reduce negative affective states but also to promote positive affective states in animals for human use. As a result, the importance of positive affective state assessment, also in mice and rats, the most widely used laboratory animal species, was noticed [21,22].

In this overview article, we present common and emerging methods to assess affective states in laboratory rodents. We focus on the implementation of these methods into applied refinement research to identify achieved progress as well as the future potential of these tools to improve animal welfare in animal-based research. In order to get an actual overview of the most frequently used methods for assessing affective states in rodents, we have illustrated the literature cited in this narrative review (Figure 1). Original research articles addressing methods for the assessment of affective states in rodents were included in the Appendix A. Protocols, guidelines, as well as articles reporting refinement strategies for analytical methods or cell culture, were excluded. We did not follow a systematic review approach but conducted a classical experts’ narrative literature review. Even though the literature research was thorough and extensive, this is an important methodological limitation of our analyses. In the following, we will address the individual methods, as listed in Figure 1.

## 2. Affective State Assessment in Laboratory Rodents

### 2.1. Physiological and Clinical Measurements

Parameters like body weight changes, general health and immune system status, reproduction success, or changes in hormone levels have been used for decades to assess the impact of husbandry and experimental procedures like caging systems, blood collection techniques, or experimentally induced disease states on the well-being of laboratory rodents (see Figure 1 for examples and references). Especially, the assessment of general health parameters and body weight measurements are nowadays implemented in virtually all score sheets for the assessment of actual or retrospective severity of experiments. Deviations in these parameters can be valuable indicators of reduced welfare in laboratory rodents. Nevertheless, no deviation is not an indication of whether an animal is experiencing a positive affective state or is in a neutral state. Even in rather negative states, for instance, in minor pain states (e.g., [23]) or some restraint stress models [24], body weight can remain stable while other behavioral parameters, for example, burrowing or wheel running, indicate a clear reduction of overall well-being. In addition, low stress hormone levels, as such, can hardly indicate a positive effect or a high level of well-being [5]. Even though these kinds of parameters played an important role in assessing the negative impact of conditions and procedures and have been essential in the refinement of animal experimentation, they are not necessarily useful for the assessment of positive welfare [5] and are, therefore, of little help in promoting positive affective states in laboratory animals.

Also, telemetric measurements of heart rate, heart rate variability, breathing rate, and body temperature have been used successfully to assess the aversive or negative impact of housing conditions (e.g., [25,26]) and experimental procedures (for a summary [27]) in laboratory rodents. The analysis of heart rate variability has especially been discussed as an indicator of positive affective states [28]. Nevertheless, we are not aware that this parameter has been successfully used to assess positive emotions in laboratory rodents yet.

### 2.2. Behavioral Measurements

The assessment of affective states based on behavioral measurements is an important tool for the evaluation of animal welfare. Approaches include the non-invasive observation of natural and spontaneously occurring behavior in the animals’ familiar environment or testing the animal under conditions that induce certain behaviors. The observation of natural behavior is of special interest for welfare assessment. One would expect that the performance of certain behaviors (e.g., grooming, nest building, play behavior) is pleasurable in itself, promotes biological functioning, and hints to a state where all other important needs of an animal are met. Other, sometimes aberrant, behaviors (e.g., overgrooming, stereotypies) may indicate reduced well-being. Nevertheless, the relative importance of many behaviors is not known yet [29], therefore linking the display of behavioral patterns with affective states of an animal is, in many cases, still speculative. In the following, the most often used behavioral approaches are discussed shortly. We cover only superficially behavioral signs that, in a broader sense, resemble clinical symptoms of reduced health, like certain aberrant movements, apathy, inactivity, disturbed circadian rhythms, or social withdrawal, while acknowledging the importance these indicators play in the routine assessment of welfare in laboratory rodents (see also Section 2.4).

#### 2.2.1. Spontaneous Behavior

It can be stated without exaggeration that the housing conditions of rats and mice can be improved from the point of view of these animals in the vast majority of animal experimental facilities worldwide. However, there are, of course, other experimental or even economic interests that stand in the way of an unrestricted improvement in husbandry conditions. In this area of conflicting interests, laboratory animals are currently frequently subject to restrictions characterized by routine, predictability, and lack of novel stimuli. These adjectives describe the human phenomenon of boredom very well, and indeed there is convincing evidence that the unpleasant affective state of boredom can also be found in animals [30]. Against this background, the spontaneous behavior of the animals in the home cage is of particular interest as it allows early warning signals to be detected as well as the evaluation of the success of refinement measures.

#### 2.2.2. Play Behavior

A promising approach for the evaluation of positive affective states is the assessment of social behaviors like play. Social play behavior, especially play fighting or rough-and-tumble play, is particularly abundant during juvenile and adolescent development in humans and animals [31,32]. As social play covers a diverse assemblage of behaviors, there exists a list of characteristics describing play behavior rather than a strict definition. Play differs structurally from purpose-bound behavior related to immediate targets. Play behavior often comprises repetitive but not stereotypical sequences that are not necessarily fully completed. Furthermore, social play involves frequent role changes, and overall play is rewarding in itself and often accompanied by specific vocalization patterns. Essentially, play behavior is usually carried out in a positive life context or is suppressed within unpleasant situations and, therefore, may indeed act as a welfare indicator [31]. For example, the structural enrichment of larger cages leads to enhanced play behavior in juvenile rats [33]. Also, the administration of certain reward eliciting drugs can stimulate social play in rats (reviewed in [32]). On the contrary, the physical restraining of male rats for 30 min lead to completely suppressed home cage play behavior in a study by Klein and co-workers [34]. However, social play behavior may also increase under certain stressful conditions, and this might be interpreted as a contradiction for play as an indicator for positive affective states [31]. Additionally, in many experimental setups analyzing play, behavior social deprivation and barren housing are used to increase the animal-experimenter bonding [31,35,36]. In mice, poor maternal care also may lead to increased play behavior in juveniles and increased anxiety related behavior later in life [37]. Thus, it becomes apparent that particular circumstances of the animal’s life history and experimental conditions have to be taken into careful consideration when play behavior in rodents is analyzed. Social play behavior can be a good indicator of welfare, but the sole assessment of play behavior seems to be not sufficient to indicate optimal or even good welfare [31]. Nevertheless, the absence of play behavior in situations that typically would be associated with play should be considered an indicator of disturbed animal welfare [6]. In this context, it is noteworthy that play behavior is rarely reported in adult laboratory mice that are housed in standard shoebox-sized cages, although it can be frequently observed when adult mice are housed in spacious conditions [38].

#### 2.2.3. Vocalization

Rats and mice vocalize in audible frequencies but also in frequencies higher than the upper audible limit of human hearing. Frequency, duration, patterns, and modulation of such vocalizations have been used to assess affective states in several species. In rats, vocalization in the range around 50 kHz is known to be indicative of positive affective states and can be elevated by hedonic stimuli, for example, play, and suppressed by aversive stimuli. It has been shown that playbacks of 50 kHz vocalizations are perceived as rewarding and neural, and pharmacological substrates elicited by these calls are consistent with human positive affective states [39]. On the other hand, in rats, 22 kHz vocalizations are indicative of negative states (e.g., after foot shock, [40]). While mice do vocalize in social contexts, examples are mating songs of male mice, social vocalization of female mice, and distress calls of mice pups [41], analyses of mouse vocalization have not revealed frequencies or patterns allowing for unambiguous discrimination between positive or negative affective states. In rats, ultrasonic vocalization has been used successfully in refinement research, for example, to prove the negative effects of gradual-fill CO_2_ killing in rats [42] or to set out the welfare enhancing effects of rat tickling (heterospecific play) (e.g., [43,44]). Contrary to rats, in mice, no vocalization indicative for negative affective states was detectable when different inhalants for euthanasia were tested [45]. However, audible vocalization as an indicator of distress was reduced by clicker training as a habituation technique in mice [46]. Ultrasonic and audible vocalization has been shown to occur in rats in acute pain (e.g., [47,48]), although the use of these markers to measure chronic pain is controversial [49,50]. In mice, less published evidence is available. While palpation induced audible vocalization in mice suffering from late-stage pancreatic cancer [51], as well as ultrasonic vocalization in chronic pain states in mice [52] are described, Williams and co-workers showed that ultrasonic vocalizations did not provide any information gain in addition to audible vocalization related to painful procedures like tail cuts conducted for genotyping [53].

#### 2.2.4. Facial Expression

Although rats and mice predominantly rely on olfactory, tactile, and auditory cues in social interaction, body posture and facial expression have been proven to be useful for welfare assessment. The detection of pain in rats and mice specifically by using the pain grimace scale [54,55] has received much attention for the use in routine pain assessment (e.g., [56,57,58]), and has been used to assess, for example, the welfare impacts of anesthesia or the severity of animal models of epilepsy (see Figure 1, e.g., [59,60]). Grimace scales have been evaluated for their suitability in detecting negative affective states, especially painful states; however, their use in refinement of handling, housing, or non-painful experimental conditions has not yet been sufficiently investigated. While grimace scales are used frequently, other facial indicators are, in contrast to their use in farm animals, rarely used for affective state assessment in laboratory rodents. An exception is the study of Finlayson and co-authors that observed a change in ear color and ear angle in rats during positive affective states, i.e., after heterospecific play [43].

#### 2.2.5. Nest Building

The construction of nests is common in most rodent species, and the motivation and ability to perform nest building behavior persists in laboratory mice and rats. Nest building performance, observed in the home cage, has proved to be a valuable and easy-to-use tool to assess brain damage or malfunction and neurodegenerative diseases but also painful and depression-like states in mice (see [61] for a summary, [62,63,64,65], more examples in Figure 1). While nest building is frequently used in mice to assess the impact of housing and experimental conditions, it is by far more rarely used in laboratory rats. One example for the successful use of nest building activity in rats for the assessment of experimental severity is described by Möller et al. [60]. The authors found a decline of nest building activity in the latter, likely more burdening, phase of an electrical kindling model. When using nest building behavior as assessment criteria, one has to consider the important thermoregulatory implications this behavior has for small rodents, especially for mice. While the reduction of nest complexity might hint to negative affective states, complex nests could, therefore, also be a result or unfavorable temperature conditions. This effect might be exacerbated, especially in single housed or sick animals that might have a need for higher ambient temperature. To our knowledge, no studies exist that analyze whether high nest building activity and nest complexity under standard animal facility temperatures and under social housing conditions are indicative of neutral or positive affective states.

#### 2.2.6. Burrowing

Mice and rats are rodent species that dig burrows under natural conditions and, similar to nest building, this behavior is also found in laboratory strains. In burrowing tests, the latency to remove items from artificial burrows or the amount removed after a certain time are analyzed (see [61] for a summary). Like nest building, the test was initially developed to monitor the progression of neurodegenerative diseases or brain lesions [66]. As burrowing activity decreases in many conditions that are likely accompanied by negative affective states, the behavior has been subsequently used in mice and rats to assess the impact and refinement of housing and experimental conditions (see Figure 1). While burrowing activity in rats is mainly used to assess different pain types or analgesic efficacy (e.g., [67,68,69]), the use of this test in mice is more heterogeneous. Burrowing behavior is known to be reduced due to post-surgical pain (e.g., [70]), experimental procedures like anesthesia [71], or experimental housing conditions like grid housing [72]. The recovery of burrowing has been used as a confirmation of the beneficial effects of experimental refinements of surgical techniques (e.g., [73]) or improvement of post-surgical housing conditions [74,75]. Additionally, burrowing has been used to monitor apathy in Parkinson’s disease models [76], stress induced anhedonia [77] and depressive-like behaviors in mouse models for major depressive disorder [78]. It is, therefore, likely that a decline of burrowing behavior is indicative of negative affective states of mice or rats. Nevertheless, most mice and rat strains show a high level of burrowing activity under standard laboratory conditions, and it is, therefore, unclear if the burrowing test could discriminate between neutral and positive affective states in rodents.

#### 2.2.7. Grooming

Another natural behavior that is used as an indicator of well-being is the grooming behavior of mice and rats. In the course of the grooming transfer test, a nontoxic, fluorescence oil is applied to the forehead or the rodent, and the speed at which the fur is cleaned is being measured. This method has been used to evaluate analgesic efficacy in mice after surgery [65]. Contrary, over-grooming has been observed in simple housing conditions compared to enriched housing [38] and has been associated with boredom and is discussed in the context of obsessive-compulsive disorders [79]. Therefore, for grooming as a welfare indicator, care should be taken not only to ensure that the behavior is displayed but also to consider patterns, frequency, and intensity.

### 2.3. Apparatus Based Behavioral Test Paradigms

In biomedical research, there are a number of well-established tests for behavioral phenotyping. These are used, for example, for testing of pharmacological treatments or for the characterization of genetically modified rodent lines [80]. In particular, some of these tests also have the potential to assess affective states. However, the short test paradigms described here have in common that the animals are taken out of their home cage and subjected to a test apparatus. Being subjected to a test situation potentially changes the affective state under scrutiny and, therefore, interpreting the results with regard to animal welfare might be difficult.

#### 2.3.1. Anxiety Related Tests

Typical tests for anxiety-related behavior are based on a conflict between approach and avoidance. Thereby, the way a test apparatus is explored provides information about the relationship between curiosity and anxiety. The most commonly used test for anxiety related behavior, the elevated plus-maze (EPM), tests exploration of shielded versus unshielded arms of an elevated cross shaped apparatus. More anxious animals tend to avoid the unprotected open arms, while more courageous animals give in to their drive to explore the entire apparatus. This test has been pharmacologically validated, showing that anxiolytic and anxiogenic drugs are indeed capable of affecting the approach/avoidance conflict accordingly [81]. Comparable conflicts for assessing anxiety related behavior are analyzed using the elevated zero maze (EZM) and the black/white box (*B*/*W* box; often also referred to as Dark-Light exploration test). Also, qualitative analysis of exploratory behavior in the well-known open field test (OF) can be interpreted with regard to anxiety related behavior by comparing the time spent in the center of the open field box to the time spent close to the wall. Paradigms summarized as free exploration tests (FET) measure the exploration of a novel environment, usually starting from the home cage. This can be realized by connecting the home cage by a tunnel to an open field arena that can be freely explored [82]. A simplified version of the FET is measuring exploration of a gridded cage top that is diagonally placed inside the home cage [71,83]. Although these tests overlap in measuring anxiety related behavior, the different test designs load differently to some extent on components influencing the affective state (e.g., exploration, novelty induced anxiety, general activity [84]). Therefore, it has been recommended to not only rely on a single short test to assess anxiety related behavior [80,82]. In addition, it should be differentiated between state and trait anxiety. While state anxiety is related to preceding events and the anxiety elicited by the test situation itself, trait anxiety rather reflects individual characteristic anxiety depending on genotype and the gene by environment trajectory that shaped the personality trait. To be more precise, tests that involve voluntary exploration (FET) of the test situation mainly measure trait anxiety while subjecting the animal forcefully to an unknown situation (EPM, EZM, *B*/*W* box, OF) rather elicits a state of anxiety. It has been shown that emotional memory is linked to state but not to trait anxiety [85]. These differences should be taken into account when relating these short tests to measuring the effects of promoting animal welfare.

All of these tests have been utilized repeatedly to answer applied research questions ranging from assessing the animal welfare impacts of housing conditions to evaluating the severity of experimental procedures in both mice and rats (see Figure 1 for examples and Appendix A).

#### 2.3.2. Preference Tests

Asking the animals themselves for their preferences is a straightforward approach in animal welfare research [86] that has been applied most frequently in studies on the refinement of housing conditions (see Figure 1 for examples). Generally speaking, preference tests utilize affective states to learn about the view and motivation of the animals tested. There are several elaborated methods available for preference testing of single or group housed animals. Some methods rely on specific test apparatuses like mazes or shuttle boxes for conditioned place preference tasks. With regard to promoting animal welfare, home-cage-based tests are considered especially meaningful as handling and habituating the animals to the test apparatuses can be influencing factors in themselves. A comprehensive overview of preference tests conducted in recent years can be found in Habedank et al. [86]. Notably, animals do not necessarily choose what is best for their health and fitness, nor can it be concluded from the preference for a certain good that the animal suffers if this item is not provided [87]. Nevertheless, preference testing as an animal centric strategy to evaluate what an animal wants will certainly be a core method in order to promote animal welfare.

#### 2.3.3. Strengths of Preferences

In most preference tests, only two items are tested against each other. It has been proposed that by a combination of multiple binary choices, a ranking can be derived [86]. However, the number of items that can be ranked against each other is usually limited. Another convincing approach to estimating the value of items chosen from an animal’s point of view is to charge a price for the decisions being made. The idea was outlined 30 years ago by Marian Dawkins, introducing consumer demand approaches into assessing animal welfare [9]. In short, these approaches test preference strength by limiting access through barriers that need to be removed or overcome [88] or by letting the animals work by means of lever pressing [89] or nose pokes in order to get access to the desired good. The amount of workload the animal is willing to “pay” is taken as a measure for the strength of the preference. Obviously, these tests are also very time-consuming and not without influencing factors. For example, the preference and the price an animal is willing to pay can be affected by the operant procedure itself. Especially when a lot of time is at hand and the animals live in otherwise impoverished housing conditions, pressing a lever in itself might be perceived as rewarding, similar to the phenomenon known as contra freeloading [90]. Overall, only a few robust data for weighted or ranked preferences are available so far.

#### 2.3.4. Cognitive Judgment Bias

As humans, we know from our own experience that subjective affective states are bidirectionally linked to cognition: decisions we make influence our emotional state and, conversely, different emotional states can lead to very different decisions. From an evolutionary point of view, this connection is adaptive because it allows, for example, to draw attention to the avoidance of predators when one is in an anxious state, while the priorities in a relaxed mood can be completely different. In contrast to affective states, cognitive abilities are much more accessible to scientific analysis and thus provide a unique opportunity to indirectly measure the underlying affective states. This was done in a very elegant way when the cognitive judgment bias (CJB) test for rats was developed by Harding et al. [91]. In brief, the CJB test relies on associative learning, combining a scalable stimulus at one end of the scale with a reward and a punishment at the other. The cognitive bias is tested after training is completed by measuring the response to ambiguous stimuli that lie between the previously learned associations. Positive affective states predispose an optimistic response (the ambiguous stimulus is more likely interpreted as if it predicts a reward), while negative affective states lead to a pessimistic expectation. Since 2004, numerous testing paradigms have been published adopting this approach for many different species. Interestingly, there were many attempts to adapt this test for mice, but, as of yet, there is no straightforward protocol that would allow repeatedly testing cognitive judgment bias in mice without facing a considerable dropout rate. Another disadvantage of the paradigm is the sometimes extremely long time it takes the animals to learn the task. In addition, the training usually includes punishment stimuli and extensive handling by the experimenter. Punishment and handling possibly change the affective state and, therefore, bias the measurement. So far, the majority of CJB tests for rats have been conducted to detect negative affective states related to an increase in pessimistic expectations (e.g., unpredictable housing [91], chronic social stress [92]). However, the CJB paradigm has a special potential to also measure positive affective states. For example, it has been shown that rats are more optimistic when provided with environmental enrichment [93], double-decker housing [94], and after being tickled [95]. Overall, CJB tests are a promising instrument for measuring affective states when methodological problems (e.g., lengthy training) can be overcome.

#### 2.3.5. Drug Self-Administration

The model of drug self-administration was developed to study addiction behavior, for example, of cocaine or heroin abuse [96]. In the last decades, the implementation of analgesic self-administration in laboratory rodents has become an interesting refinement aspect. The strategies include self-administration via drinking water, oral route via nut paste, or aqueous gel or intravenous/intrathecal self-administration via catheter [97,98,99,100,101]. However, monitoring the correct dosage is more difficult and an injection of the analgesic agent may be more effective in some cases [97]. Overall, drug self-administration offers the possibility of moving from a poor welfare condition to a better one. However, little is known about positive affective states other than the states of intoxication perceived as positive in addiction research.

### 2.4. Scoring Systems, Score Sheets, Composite Scores

The European directive 2010/63/EU as well as the Swiss legislation request the assessment of the actual or retrospective severity of procedures in animal experiments. The most common method for documenting the severity level that experimental animals have actually experienced is the use of score sheets specific to the experiment. Score sheets have been recommended to include specific and sensitive assessment parameters, which can be behavioral and clinical indicators of animal health status, for example, the appearance of the animal including the condition of the fur and body orifices, as well as changes in spontaneous and provoked behaviors of the animal. In addition, score sheets should, whenever possible, also include experiment-specific parameters [102,103]. Scoring systems often include certain behaviors that are typical for pain in rodents in order to assess postoperative pain or to implement early endpoints in various disease models [104,105,106]. However, in most cases, the selected behaviors and symptoms are unlikely to reflect the emotional component of pain but are simply the physical reaction to painful stimulus (e.g., signs like twitch, back arch, stagger) [104,107]. Again, the mere absence of behaviors typical for pain is by no means a sign of good welfare of the animal. Typical score sheets are designed as tools for the assessment of the general health condition of an animal and to secure timely application of measures to reduce burden and to apply humane endpoints. However, they do not seem suitable for the determination of positive affective states. Nevertheless, development of easy-to-assess indicators of affective state and their implementation in the obligatory standard monitoring of laboratory rodents would benefit animal welfare in routine laboratory and experimental settings significantly.

### 2.5. Automation of Assessment

The automation of methods for the assessment of affective states in rodents is an emerging field to facilitate observation, to increase throughput, and to reduce the putative influence of the experimenter. Especially in the area of home-cage tracking and the detection of changes in facial expression, considerable progress has been made in recent years [108,109,110,111].

Home-cage activity is a sensitive measure for detecting aberrant behavior, especially with regard to the time course and long-term observations. Manual scoring of home cage behavior is, however, laborious and very time consuming, therefore there is a high demand for automated systems. For example, activity measures can be derived from running wheels and can serve as an indicator of the severity of experimental procedures [24].

Commercially available systems (e.g., Noldus, TSE, or Tecniplast) promise to enable an automated home-cage observation to analyze the whole day activity of grouped housed individuals [112,113,114,115]. Some systems are best suited for single cage monitoring, but others are capable of monitoring entire rodent racks, such as the rodent Big Brother Project [116]. However, these systems often lack automated, quantitative, and accurate assessment of social interaction. In recent years, more and more freeware has become available to address these problems, for example, AnimApp, M-Track, [108,109,117,118]. In addition, vital parameters, like body temperature, drinking, and eating events, can be obtained. Such systems have been used successfully, for example, to identify postoperative pain [119].

The integration of artificial intelligence using novel deep learning approaches for behavioral observation and data analysis is a promising approach. [108,109]. In the research area of computer vision and learning, such models potentially reveal structures in mouse behavior free of any observer bias. Using these methods yield new possibilities to assess also subtle changes in behavior [120]. In connection with the development of deep learning approaches, for example, the software “deep squeak” was developed to analyze the vocalization of rodents [121]. Additionally, the automated recording of the pain face of mice and rats has also been advanced in recent years. The integration of deep learning and rodent face finder enables a semiautomated and non-human-biased evaluation of the grimace scale [55,110,111,122,123,124]. For black mice, 99% accuracy can be achieved [110].

## 3. Conclusions

In this overview article, we presented common and novel methods for assessing affective states in laboratory rodents. Most of the applied approaches focus on capturing short-lasting affective states often in direct relation to experimental procedures. However, the promotion and evaluation of animal welfare should be understood as an approach that concerns the whole life of an animal. It would, therefore, be desirable to monitor affective states over long periods of time. Thus, long-term observations and assessment criteria for sustainable improvement should be further promoted. Many of the measures that we reviewed relate to assessing negative affective states (e.g., pain grimace scale, loss of body weight), and often, the absence of obvious indicators for negative affective states is interpreted as being positive. This, however, falls short in truly assessing positive affective states and can be seen as a lost opportunity in current welfare and refinement research in laboratory animal science. Moreover, some of the indicators for positive affective states found in the literature (Figure 1) rather indicate functioning ‘normal’ behavior than necessarily being indicative of positive affective states (e.g., grooming, nest building, burrowing). A recent analysis of literature on positive animal welfare in farm animals finds the body of literature on this topic to be relatively small [125]; we see the same in laboratory animal science, indicating that still much more research is required in this field. As visualized in Figure 1, most of the measures are thought to capture the extremes of the scale from negative to positive valence. Especially for the range between neutral and positive valence, there is a lack of established measures. Here, new arising methods for the automated assessment of behavior and facial expressions potentially will help to fill this gap in evaluating affective states. It is obvious that the analysis of behavior plays an increasing role in rodent welfare assessment. Changes in the display of social play and ultrasonic vocalization are promising tools for the evaluation of positive affective states, especially in rats. Precisely because ultrasound voices and play behavior are less conspicuous in mice, there is a need to refine the assessment of spontaneous behavior in order to find reliable indicators of positive affective states in laboratory mice. Here, new approaches using Computer Vision, together with Deep Learning algorithms to routinely monitor daily behavior, will potentially lead to a quantum leap in refinement research.

Nevertheless, we are not there yet, and “Minimizing harm, maximizing good” is an obligation for animal users. Consequently, the absence of positive affective states constitutes a welfare problem [126]. We are convinced that a shift from standard assessment of negative affective states towards positive affective state assessment, and the growing significance of the concept of positive animal welfare in laboratory animals will be of benefit for all animals used in animal experiments and in animal experiments performed for refinement research. Facilitating the use of validated, feasible, and robust tools to assess affective states, both positive and negative, is, therefore, a paramount objective in the field of animal experimentation.

## Figures and Tables

**Figure 1 animals-09-01026-f001:**
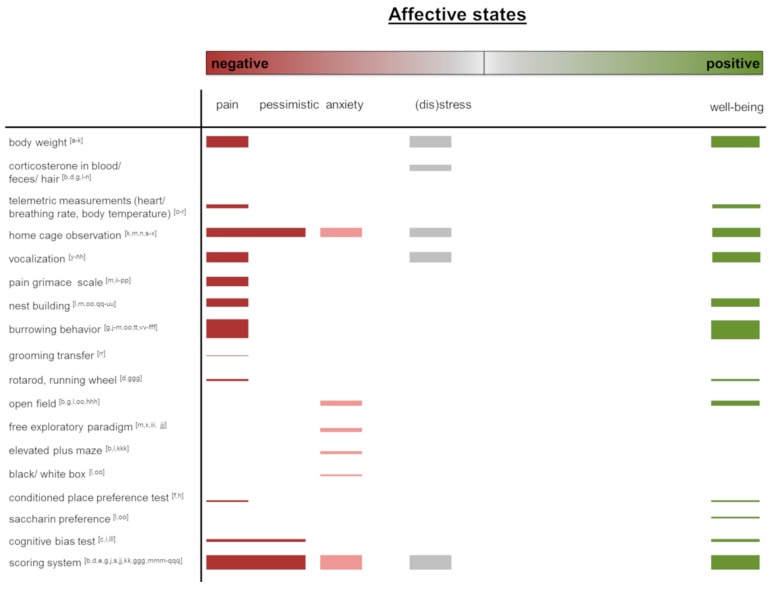
Graphical overview of the methods assessing positive or negative affective states as described in this article. Original research articles addressing methods for the assessment of affective states were included in the figure. Protocols, guidelines, as well as articles reporting refinement strategies for analytical methods or cell culture, were excluded. The thickness of bars represents the number of publications. The publication list is available in the Appendix A. The order and assignment of the measurements for negative or positive affective states assessment were made by the authors.

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
