# Peer review of "Assessing Affective State in Laboratory Rodents to Promote Animal Welfare—What Is the Progress in Applied Refinement Research?"

_animals, 2019, doi:10.3390/ani9121026_

Round 1

Reviewer 1 Report

I have some serious problems with this paper.

1 The review method is not state of the art: not systematic or systemised and not transparant in its methods. The search terms are very limited in the sense that the lack any synonymes and do not seem to cover the question: The authors search for methods to assesses affective states in mice and rats. E.g. a publication about pain behavior in mice would be missed. No wonder they only found 32. Most of the literature they use in their review comes from "selected literature". What is that, how selected? Ample opportunity for bias! These flaws undermine the value of this review.

2. The introduction has to many unclear and needless complicated sentences and words and is therefore often hard to understand (e.g. line 60-65, the definition in lines 67-69, 89-91). Moreover it is unclear what the used definition of welfare comprises (the reference refers to a publication "in press") and it is unclear why this definition is mentioned while it does not return in the rest of the text. Further more the reason given to use the term "affective state" is a little bit strange: it is to be as unbiased as possible, while in the previous sentence the term is mentioned as having a strong athropomorphic connonation?? To be short, the introduction need rigorous editing using clear language and skipping unnecessary items.

3 The paper misses focus. While the title and introduction suggest a focus on positive welfare, the review is about assessing welfare methods in general. In the description of some methods positive welfare is not even mentioned. The figure 1 is for me not fully understandable. Also the conclusion could be sharper, it seems to me that on basis of the review we must conclude that positive welfare could be measured with some methods but that in practice it is not done. In the title also "refinement"is mentioned. If the authors want to focus on refinement the should elaborate more on how positive welfare can refine animal experiments (for example does it compensate for negative welfare..). In this text refinement does not get enough attention.

4 The described results of the the review are still interesting, but are for me compromised by my doubts about the review method (point 1)  

Author Response

Dear reviewer,

Thank you very much for the helpful remarks about our manuscript. Based on your comments and those of the second reviewer we revised the introduction and conclusion section significantly. By this we have focused the statements of the manuscript and also emphasized the message more strongly in the appropriate place. In the following, we will comment on each of the points that were raised.

I have some serious problems with this paper:

The review method is not state of the art: not systematic or systemised and not transparant in its methods. The search terms are very limited in the sense that the lack any synonymes and do not seem to cover the question: The authors search for methods to assesses affective states in mice and rats. E.g. a publication about pain behavior in mice would be missed. No wonder they only found 32. Most of the literature they use in their review comes from "selected literature". What is that, how selected? Ample opportunity for bias! These flaws undermine the value of this review.

Author response: We agree with the referee that this manuscript is not a systematic review. It is an expert opinion on the status quo of the applied use of affective state assessment in rodents. By no means, we intended to mislead the reader. We have added the following information in the introduction: “We therefore did not follow a systematic review approach but conducted a classical experts`, narrative literature review. Even though the literature research was thorough and extensive, this is an important methodological limitation of our analyses.”

In addition, we revised the description of the figure legend as well as the description of the figure in the introduction section to remove any misleading formulation. Also please note that our reference lists contains >130 references plus the supplemental references used for compiling Fig 1. Thus, the text is rather based on our selection of the relevant literature while the figure summarizes the situation based on a literature search using the outlined search strategy. Nevertheless, in our view, the figure illustrates the current situation very well. 

The introduction has to many unclear and needless complicated sentences and words and is therefore often hard to understand (e.g. line 60-65, the definition in lines 67-69, 89-91). Moreover it is unclear what the used definition of welfare comprises (the reference refers to a publication "in press") and it is unclear why this definition is mentioned while it does not return in the rest of the text. Further more the reason given to use the term "affective state" is a little bit strange: it is to be as unbiased as possible, while in the previous sentence the term is mentioned as having a strong athropomorphic connonation?? To be short, the introduction need rigorous editing using clear language and skipping unnecessary items.

Author response: We thank the reviewer for this important remark and streamlined the introduction significantly to make this part more clear and comprehensible.

The paper misses focus. While the title and introduction suggest a focus on positive welfare, the review is about assessing welfare methods in general. In the description of some methods positive welfare is not even mentioned.

Author response: We apologize for any seemingly misleading wording. First of all, we did not aim to solely focus on positive affective state assessment. Therefore, we used just the wording “affective state assessment” in the title. Furthermore, some methods only allow the assessment of a negative affective state. The absence of a negative affective state does not necessarily mean that the animal is well. Therefore, we revised the description of our methods to state clearly if the assessment of a positive or only a negative affective state is possible.

The figure 1 is for me not fully understandable.

Author response: Thank you for this important comment. We revised the figure including the figure legend to make it more comprehensible. 

Also the conclusion could be sharper, it seems to me that on basis of the review we must conclude that positive welfare could be measured with some methods but that in practice it is not done.

Author response: Actually, we propose affective state assessment as a tool to improve animal welfare. Only if positive as well as negative affective states can be measured, it is possible to analyse the impact on animal welfare that certain housing conditions or experimental procedures have. This is in our opinion the basis of every approach to refine animal use in science. We agree that while negative state assessment is relatively common in laboratory rodents, positive affective state assessment is still rare. In our view this is a lost potential for the field of refinement research. We added a statement to the conclusion.

In the title also "refinement"is mentioned. If the authors want to focus on refinement the should elaborate more on how positive welfare can refine animal experiments (for example does it compensate for negative welfare..). In this text refinement does not get enough attention.

Author response: We thank the reviewer for pointing to this. The assessment of either negative or positive affective state is the basis of every approach to refine animal use in science. We feel that every evidence based recommendation for refined procedures may improve animal welfare. If the promotion of positive affective states can counterbalance negative affective states (induced by procedures) is an important question but outside the focus of our manuscript. The scientific purposes in animal science are manifold and do not necessarily have to have the refinement as a goal but have integrated it to achieve better results. Hence, we have included also articles that deal with the development of methods, but have not yet been used in applied refinement research or that have implemented methods of affective states assessment without directly dealing with refinement.

The described results of the the review are still interesting, but are for me compromised by my doubts about the review method (point 1)  

Author response: We thank the reviewer for the constructive comments and hope that our revised manuscript is now more clear, comprehensible and acceptable for publication.

Reviewer 2 Report

I was a bit confused about the content of lines 66-67. Basically, the au. seems to suggest that we have access only to current temporal slices and also speaks of inferences from this only to future welfare states. Why doesn't the access to current time slices allow us to characterize present welfare states. Indeed, wouldn't this be less problematic?

Albeit, somewhat tangential to the paper, I have 2 questions that naturally arise from reading the paper:

    1. Fundamentally, how do even aversive affective states get grounded in behavioral and physiological patterns? (Wouldn't one to be able, at first, to identify aversive states independently of such patterns?

    2. What are the au.'s views on the ethics of purposely supplying stimuli that cause aversive mental states to rodents?

Author Response

Dear reviewer,

Thank you very much for the helpful remarks about our manuscript. Based on your comments and those of the second reviewer we revised the introduction and conclusion section significantly. By this we have focused the statements of the manuscript and also emphasized the message more strongly in the appropriate place. In the following, we will comment on each of the points that were raised.

I was a bit confused about the content of lines 66-67. Basically, the au. seems to suggest that we have access only to current temporal slices and also speaks of inferences from this only to future welfare states. Why doesn't the access to current time slices allow us to characterize present welfare states. Indeed, wouldn't this be less problematic?

Author response: We agree with the reviewer that this part of the introduction might be misleading. As these sentences are not necessary to understand the main aim of our manuscript, we omitted this part of the introduction.

Albeit, somewhat tangential to the paper, I have 2 questions that naturally arise from reading the paper:

Fundamentally, how do even aversive affective states get grounded in behavioral and physiological patterns? (Wouldn't one to be able, at first, to identify aversive states independently of such patterns?

Author response: We thank the reviewer for this remark. Overall, the assessment of either negative or positive affective states is anything but trivial. At the moment, two different strategies are pursued: the assessment of a single state (e.g., pain face, body weight) and the assessment of complex behavior pattern (e.g., play behavior, exploratory behavior). Both strategies have their justification. However, the assessment of a single state independently of complex pattern is often not sufficient as stated in the manuscript. Therefore, the observation of complex non human biased behavior patterns are used more and more frequently. As a result of our review, a combination of both strategies would be favourable to get a realistic overview of the actual affective state of a certain animal. Based on your comment, we revised the introduction as well as the conclusion section to make this important point more comprehensible.   

What are the au.'s views on the ethics of purposely supplying stimuli that cause aversive mental states to rodents?

Author response: Thank you for raising this question. It is an inherent ethical dilemma of refinement research. Less favourable, mostly standard, conditions and procedures are used as control groups for the comparison with less aversive and refined conditions and procedures. As with every animal experiment this kind of studies are only justifiable if the expected benefit for the animals that will undergo the refined procedures in the future will outweigh the welfare costs of the animals used for the refinement research. It is unlikely that this is always the case. The authors are convinced that a shift towards more interest and awareness of the occurrence of positive affective state, methods to assess positive states and the growing significance of the concept of positive animal welfare also in laboratory animals will be of benefit of all animals used in animal experiments and in animal experiments performed for refinement research. We added a statement in the conclusion.

Round 2

Reviewer 1 Report

In the letter and the new text my previous critical remark are met. The introduction is clear now, and matches the rest of the article. I have no further remark (although I still would prefer a more sophisticated literature search!)